# The Quality of Life in Elderly Patients in Comprehensive Conservative Management or Hemodialysis: A Case–Control Study in Analogous Basal Conditions

**DOI:** 10.3390/nu16173037

**Published:** 2024-09-09

**Authors:** Francesca K. Martino, Daniela Campo, Lucia Federica Stefanelli, Alessandra Zattarin, Daria Piccolo, Martina Cacciapuoti, Marco Bogo, Dorella Del Prete, Federico Nalesso, Lorenzo A. Calò

**Affiliations:** 1Nephrology, Dialysis and Transplantation Unit, Department of Medicine (DIMED), University of Padova, 35128 Padua, Italy; danielacampo92@gmail.com (D.C.); luciafederica.stefanelli@unipd.it (L.F.S.); martina.cacciapuoti@studenti.unipd.it (M.C.); marco.bogo@studenti.unipd.it (M.B.); dorella.delprete@unipd.it (D.D.P.); federico.nalesso@unipd.it (F.N.); 2Department of Medicine (DIMED), Clinical Nutrition, University of Padua, 35128 Padua, Italy; alessandra.zattarin@aopd.veneto.it (A.Z.); daria.piccolo@aopd.veneto.it (D.P.)

**Keywords:** quality of life, conservative management, hemodialysis, elderly, pain

## Abstract

Background/Objectives: Comprehensive conservative management (CCM) is a viable treatment option for elderly patients with end-stage kidney disease (ESKD). However, it involves a significant change in dietary habits, such as adopting a low-protein diet. Therefore, it is crucial to understand its impact on the patient’s quality of life (QoL), particularly when compared to hemodialysis (HD). The study aims to evaluate the differences in the QoL between patients undergoing CCM and HD. Methods: The study included 50 patients over 75 with ESKD, with 25 patients in the CCM group and 25 in the HD group. The CCM group followed a personalized low-protein diet, while the HD group did not have protein restrictions. Various parameters were assessed, including demographic data, urine output, blood tests, comorbidity index, Visual Analog Scale (VAS), and hospitalization. The SF-12 questionnaire assessed the QoL, and the Physical Composite Score (PCS) and Mental Composite Score (MCS) were calculated. Results: The study revealed no age and comorbidity index differences between CCM and HD patients. In contrast, CCM patients reported significantly better physical and mental well-being than HD patients. In univariate analysis, CCM (B 0.24, *p* = 0.001), protein intake (B −0.004, *p* = 0.008), hospitalization (B −0.18, *p* = 0.024), urine output (B 0.25, *p* = 0.001), and VAS (B −0.26, *p* < 0.001) influenced the PCS. At the same time, only the type of treatment (B = 0.15, *p* = 0.048), urine output (B 0.18, *p* = 0.02), and VAS (B −0.14, *p* = 0.048) influence the MCS. In contrast, in multivariate analysis, only CCM contributed to an improved PCS (B 0.19, *p* = 0.003) and MCS (B 0.16, *p* = 0.03), while a higher VAS worsened the PCS (B −0.24, *p* < 0.001) and MCS (B −0.157, *p* = 0.0024). Conclusions: In elderly patients with similar basal conditions, health-related QoL perception is better in CCM than in HD patients.

## 1. Introduction

The prevalence of end-stage chronic kidney disease (ESKD) has been increasing among elderly patients. In 2020, the United States Renal Data System estimated an incidence of ESKD in 1447 per million population (pmp) among individuals aged 75 years and older, which is about three times the incidence in patients aged 45–64 years (598 pmp) [1]. Elderly patients have a higher prevalence of comorbidities [2], such as diabetes mellitus, hypertension, congestive heart failure, clinical depression, and a history of ischemic heart disease, which contribute further to increasing their frailty. Heart failure, dementia, peripheral vascular disease, and decreased albumin seem to be highly associated with cardiovascular mortality within the first three to six months after the initiation of hemodialysis (HD) [3,4,5,6]. In ESKD patients, non-traditional risk factors such as hyperkalemia and calcium–phosphate metabolism impairment increase the risk of complications [7,8,9,10,11] and worsen the elderly patient’s condition, making dialysis procedures doubtful to improve survival [7].

In older people, HD treatment could lead to a lower quality of life due to the need to regularly visit a hemodialysis center and the psychological burden of dialysis dependence, as well as potential complications such as hypotension [12], cramps [13], acute heart failure [14,15], pulmonary edema [15], thrombosis, and infection [16,17]. For elderly patients with preserved urine output, conservative management might be the preferred approach for different reasons. First, ESKD patients are not burdened by HD complications, i.e., hypotension, cramps, and vascular access complications. Second, it permits the elderly to maintain their habits without the need to access hospital facilities. Third, it effectively controls uremia complications by improving uremic toxin retention, anemia, and blood pressure, even in ESKD [18,19]. Finally, conservative management can slow the progression of glomerular filtration loss [20,21]. In Table 1, all the potential benefits of CCM in elderly ESKD patients are outlined.

However, choosing the best ESKD treatment remains a challenge for nephrologists. The lack of conclusive evidence proving that CCM is comparable to RRT in patient survival, insufficient resources for CCM diffusion, and the necessity of strict adherence to dietary and drug treatments represent the prominent barriers to CCM expansion [7]. Furthermore, a low-protein diet requires significant changes in eating habits, impacting the patient’s quality of life (QoL). In ESKD, protein restrictions could reach 0.3–0.4 g/kg/day, necessitating low-protein foods and keto-analogues integration, which could be challenging for elderly patients.

Currently, only some heterogeneous studies [22,23,24,25,26,27] explored the difference in the QoL between conservative management and HD in elderly patients with inconclusive results. Specifically, most of the studies [23,25,26,27] had significant differences in age and/or comorbidity between cases and controls. Only a few studies have investigated the possible predictors of the QoL [23,24,28]. These issues limit the reliability of the singular result and do not allow a clear message about the QoL in ESKD elderly patients. In this context, the actual impact of each treatment on the QoL needs to be clarified. This study aims to evaluate the QoL in elderly ESKD patients with comparable basal conditions through the perception of the patients with a QoL questionnaire. Secondly, it aims to estimate the relationship between the perception of the QoL and general and specific ESKD conditions in the first year of treatment.

## 2. Materials and Methods

We conducted a retrospective case–control study at the Nephrology, Dialysis, and Kidney Transplantation Unit of the University of Padua Hospital. We included the first 25 elderly patients who initiated CCM or HD between 1 April 2022 and 31 March 2023. The CCM program commenced on 30 September 2022. Consequently, patients who developed ESKD before September and chose HD were enrolled in the HD group, while the patients who reached ESKD after September 2022 and chose CCM were enrolled in the CCM group if inclusion and exclusion criteria were satisfied.

The inclusion criteria were as follows:-Age over 75 years;-eGFR < 12 mL/min/1.73m^2^;-6 months of CCM or HD;-Availability of a 12-item Short-Form Health Survey at the end of the observational period.

The exclusion criterion was a refusal to participate in the study.

All patients had six months of observational period.

Type of intervention

Patients who chose HD treatment commenced treatment based on their clinical condition and KDIGO guidelines, considering the glomerular filtration rate and uremia-related symptoms such as asthenia, nausea, vomiting, sodium–potassium imbalance, calcium–phosphate metabolism, and fluid overload. All HD patients received standard care with 3 to 4 h of HD treatment three times a week and blood examinations at least once a month. Regarding the diet, HD patients did not have protein restriction, but they might have had suggestions about phosphate and potassium intake based on their blood examinations.

Patients who opted for CCM immediately revised their low-protein diet. If they were at low risk of malnutrition and had no swallow impairment, they received a very low-protein diet (0.4 g/kg/day of protein) supported by keto-analogues. If they were at moderate-to-high risk of malnutrition or could not take keto-analogues, they started or continued with a low-protein diet, according to the nephrologist’s judgment. Dieticians planned a personalized diet for these patients, including a controlled amount of phosphate and potassium, taking into account the nephrologist’s advice and the patient’s eating habits.

According to nephrologists’ advice, both HD and CCM received all supporting treatments for CKD (erythropoietin, potassium binder, phosphate binder, Vitamin D analogs, calcium mimetics, sodium bicarbonate, diuretics, anti-hypertensive treatment, and antihistamine treatment).

For each patient, we collected the following parameters:Demographic data as the gender and age.The Charlson Comorbidity Index (CCI) adjusted by the patient’s age (+3 in patients between 70 and 79 years old, +4 in patients over 80 years old) was detected at the beginning of HD or CM. Specifically, CCI evaluates the following conditions: diabetes, congestive heart failure, peripheral vascular disease, chronic pulmonary disease, liver disease, hemiplegia, renal disease, hematological or metastatic cancer, and acquired immunodeficiency syndrome.Blood examinations: creatinine, urea, sodium, potassium, calcium, phosphorus, parathyroid hormone (PTH), Vitamin D 25-OH, bicarbonate, albumin, cholesterol, and triglycerides. In all patients, venous blood samples were collected between 7 and 8 am at the end of the observational period. In HD patients, samples were collected in the long interval before the HD session.Urine output in 24 h.The median Visual Analog Scale (VAS) was calculated for pain assessment for each patient according to the scoring reported in the clinical notes in the observational period. To avoid differences in the number of observations between the case and control groups, we considered all outpatient visits in CCM patients and the first evaluation of the month in HD patients.Severe complications such as sepsis, pneumonia, myocardial infarction, and cerebrovascular events that required hospitalization were detected during the observational period.

### 2.1. Outcomes

The study’s primary outcome was to assess the differences in the QoL between CM and HD patients. To weigh the impact of treatment on QoL, we administered the Italian version of the 12-item Short-Form Health Survey (SF-12) to each patient at the end of the observational period. SF-12 is a self-reported questionnaire that evaluates the impact of health status on QoL, considering different features, such as limitations in physical activities, social activities, and usual role activities related to health problems, limitations due to emotional issues, pain and fatigue, and general health perception (Appendix A). The results were summarized in the Physical Composite Score (PCS) and the Mental Composite Score (MCS), which can score between 0 and 100, where 0 represents illness and 100 well-being.

The secondary outcome was to assess how the clinical conditions (age, CCI, ESKD management, uremia control, pain perception, severe complications, and hospitalization) could impact QoL.

### 2.2. Sample Size

We estimated the sample size of the study using Kelsey’s formula by the OpenEpi calculator (https://www.openepi.com/SampleSize, accessed on 1 March 2023), comparing the proportion of patients who achieved at least 50 scores in SF-6D with a power of 80%, confidence interval (2-sided) 95%, and the ratio of sample size equal to 1. Specifically, considering the date of publication and the type of population, we used the results of Shah K.K. et al. [22], obtaining a sample size of 25 patients for each group. To avoid selection bias, we considered the first 25 patients who satisfied the inclusion criteria in the enrollment period for each arm.

### 2.3. Statistical Analysis

Categorical variables were described as percentages. Normally distributed continuous variables were reported as mean ± standard deviation, whereas non-normally distributed variables were reported as median with interquartile range. The normality of each variable was evaluated using the Kolmogorov–Smirnov test. Student’s *t*-test, Mann–Whitney U, and the chi-square test were used to compare continuous and categorical variables as appropriate. Spearman rho was used to detect the correlation between variables. Univariate and multivariate linear regressions were evaluated to detect the predictors of PCS and MCS. All continuous variables that were non-normally distributed were normalized by natural log transformation. Statistical significance was assessed by a two-tailed test with a *p* < 0.05. Statistical analysis was performed with the IBM Corp. Released 2021. IBM SPSS Statistics for Windows, Version 28.0. Armonk, NY: IBM Corp.

## 3. Results

Fifty patients with a mean age of 81 ± 4.5 were enrolled.

Three CCM patients followed a low-protein diet (0.6 g/kg/day), and 22 followed a very low-protein diet (0.4/kg/day) supplemented by keto-analogues. In contrast, all HD patients followed a free diet. Specifically, dialysis and CCM patients were almost comparable at the beginning of treatment. However, CCM patients were more likely to be female (56% versus 24% *p* = 0.02, respectively) and had higher levels of hemoglobin (11 versus 10.6 *p* = 0.002, respectively), as reported in Table 2.

At the end of the observational period, CCM patients had lower levels of urea (*p* = 0.01), potassium (*p* = 0.01), and phosphate (*p* = 0.03). As expected, they had higher urine output (*p* = 0.001), as reported in Table 3.

During the six-month observational period, CCM patients had a trend toward a lower hospitalization rate (52% vs. 72%, *p* = 0.16), as reported in Figure 1.

Furthermore, as reported in Figure 2, CCM patients had a significantly lower incidence of pneumonia (16% in HD patients versus 0% in CCM patients, *p* = 0.01) and sepsis (28% in HD patients versus 0% in CCM patients, *p* = 0.001). In the same period, HD patients showed a trend toward a higher incidence of myocardial infarction (12% in HD patients versus 0% in CCM patients, *p* = 0.07) but a similar incidence of cerebrovascular events (4% in HD patients versus 0% in CCM patients, *p* = 0.31).

The PCS and the MCS were highly correlated with a Spearman rho equal to 0.52 (*p* < 0.001). According to physical and mental scoring, the SF-12 questionnaire results for the entire population are reported in Figure 3. Specifically, CCM patients had a significantly higher PCS and MCS compared to the HD patients, with a median of 31.4 [27.8–38.4] versus 44.7 [37.2–51.5] (*p* < 0.001) and of 40.7 [34.6–51.7] versus 52.9 [43.9–57.6] (*p* = 0.024), respectively.

In our series, the PCS is significantly influenced by the type of treatment (B 0.24, *p* = 0.001), protein intake (B −0.04, *p* = 0.008), urine output (B 0.25, *p* = 0.001), the absence of hospital admittance, and the VAS (B 0.18, *p* = 0.012) in univariate analysis. In multivariate analysis, only the ESKD management (B 0.19, *p* = 0.003) and VAS (B −0.24, *p* < 0.001) were independent predictors of the physical score. Contextually, the MCS was significantly influenced by the type of treatment (B 0.15, *p* = 0.048), urine output (B 0.18, *p* = 0.02), and the VAS (B −0.14, *p* = 0.048) in univariate analysis. In multivariate analysis, ESKD management (B 0.16, *p* = 0.03) and pain perception (B −0.157 *p* < 0.001) were the only independent predictors of the mental scores.

Table 4 and Table 5 summarized univariate and multivariate analyses for the prediction of the PCS and MCS.

## 4. Discussion

Our study compared the QoL between patients with similar age, comorbidity index, and overall health receiving either CCM or HD. Our findings revealed that CCM patients experienced significantly better physical and mental well-being than HD patients. Furthermore, they had lower rates of severe infections, a better control of urea retention, sodium, potassium, and phosphate levels, and higher urine output. In our analysis, physical well-being was influenced by CCM, protein intake, urine output, pain perception, and hospital admissions. At the same time, mental health was influenced by the type of ESKD management, urine output, and VAS. Our multivariate model showed that CCM and the VAS were independent predictors of both physical and mental health according to the SF-12 questionnaire. To our knowledge, the present study is the first to compare CCM and HD patients with similar baseline age, comorbidity index, and overall health.

Our results support previous findings by Shah K.K. et al. [22]. However, we also provide additional insights by identifying factors associated with better physical and mental well-being in CCM compared to RRT. Specifically, they found better scores in the SF-36 regarding the burden of disease and a trend in significant differences for the PCS (*p* = 0.12) and MCS (*p* = 0.16) in the two groups. The lack of comorbidity assessment and the inclusion of peritoneal dialysis patients in the control group might justify the little discrepancy in the results. Three studies did not show differences in the QoL between conservative management and RRT [23,24,26]. The Verberne et al. study [23] had a prominent difference in age. The Iyasere O. et al. study [24] significantly differed in comorbidities. Finally, Seow et al. [26], despite the lack of difference between CCM and RRT in health-related QoL, showed a slower decline in the PCS over time and a higher PCS and MCS at 12 months in the CCM group, findings which move in the same direction as our study. The extended control group to peritoneal dialysis and the different basal conditions could explain the differing results compared with ours. It is not surprising that older patients with higher comorbidity burden may experience a worse QoL. Furthermore, including a home RRT in the control group could dilute the differences in the QoL. Only van Loon et al. [25] observed a slight but significant decline in the QoL among CCM patients. In this study, the CCM patients were older and had lower baseline hemoglobin and albumin than the dialysis patients, which likely impacted the QoL of CCM patients. Our study compared the QoL in HD and CCM patients with similar baseline conditions, which helped to uncover the difference in the QoL between the two groups, and it is one of the strengths of our report.

Some interesting hypotheses can explain our findings. The lower need for access to the hospital could allow better health perception and, consequentially, justify a higher mental scoring. In contrast, we found a more prominent impact in the PCS, which seems to evocate a worse physical condition in the HD group, likely related to the reduced tolerance to the HD of elderly people. It is known that elderly patients have a higher rate of HD side effects related to cardiovascular instability compared to young patients [29]. Intradialytic hypotension remains an important issue that is related to ultrafiltration rate [29,30,31], diabetes [29,32], and chronic heart disease [33,34,35], but it also impacts the QoL [36,37] and survival [38,39] of HD patients. Specifically, intradialytic hypotension and related symptoms such as nausea, cramps, asthenia, and dizziness could be challenging to overcome, especially in elderly patients, and they can compromise the perception of health status.

We found a lower rate of severe infective complications in CCM. These issues should be related to higher access to hospital facilities and HD central catheter/arteriovenous fistula infection. However, it could also be due to a better control of uremic syndrome with a lower impairment of immune function [40,41]. Currently, there is insufficient evidence to assess residual renal function’s possible role in uremic toxins and the immunity system. We plan to explore this issue in future studies to define the role of diet in treating ESKD.

Stimulatingly, we found a significant negative relationship between protein intake and the PCS in univariate linear regression, which suggests how protein restriction could negatively influence the perception of physical well-being. Furthermore, this interference also represents a possible issue in adherence to a very low protein diet [42]. Previous studies showed no issues with nutritional status in personalized, very-low-protein diets [43,44]. However, the current results suggest that the higher restriction in protein intake should be managed carefully, especially in underweight or near-to-underweight people who have absolute lower protein intake. Previous studies have shown that the hospital admission rate, especially for severe events, seems to impact the QoL [45,46]. Therefore, our results on the PCS in univariate analysis move in the same direction, showing how hospital admissions worsen the physical sphere scoring. Finally, we highlighted the importance of pain scoring in the QoL in elderly ESKD patients. Pain is a common symptom in ESKD, with a prevalence that can reach 50% of patients, and it is often undertreated [47], especially in CCM patients in whom NSAIDs are forbidden while opioids can worsen constipation. Obtaining pain control and optimized pharmacological treatment is not simple in this context. The lack of adequate chronic pain treatment explains the relevant role of pain in the QoL in our cohort.

Our study has some limitations. First and foremost, those associated with the retrospective study design. Second, the low number of patients for linear regression analysis can limit the reliability of our analysis, reducing the actual effect of other covariates in the model. However, we estimated the sample size according to the study’s primary aim. Furthermore, the previous studies [23,24,27] that explored the predictors of the QoL were heterogeneous regarding the type of dialysis and instruments used to assess the QoL, limiting the evaluation of the sample size for the secondary aim. We are planning a new analysis to confirm our findings. Third, the VAS could not be adequate for chronic pain scoring, but the retrospective nature of our report did not allow any different tool to assess the pain [48]. Fourth, we did not take into account the utilization of pharmacological treatment for anxiety and depression, which could potentially interfere with the study’s results. Finally, we had a significant difference in gender distribution, which theoretically could impact the results. In a previous study, the female gender was related to a higher inclination to choose conservative management [49]. Our results move in the same direction, but understanding the reasons for this difference and its impact on the results is challenging.

## 5. Conclusions

Our study showed for the first time how the CCM of ESKD in elderly patients with similar basal conditions impacts the QoL more favorably than HD. Specifically, CCM improves the PCS and MCS in the 12-month follow-up period. Furthermore, the absence of hospitalization, higher urine output, better pain control, and higher protein intake improves the PCS in the follow-up period. In contrast, only CCM, higher urine output, and better pain control were related to a higher MCS. Finally, the discrepancy between our and other results highlights how basal conditions can impact the perception of health-related QoL. Thus, in elderly patients, assessing a better perception of the QoL without dialysis endorses CCM, when possible, as the best option for them.

## Figures and Tables

**Figure 1 nutrients-16-03037-f001:**
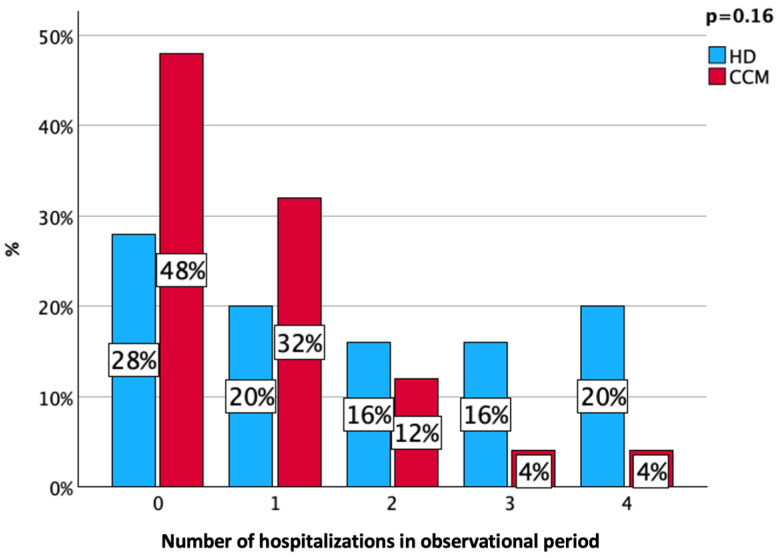
Hospitalization rate in the follow-up period according to ESKD treatment.

**Figure 2 nutrients-16-03037-f002:**
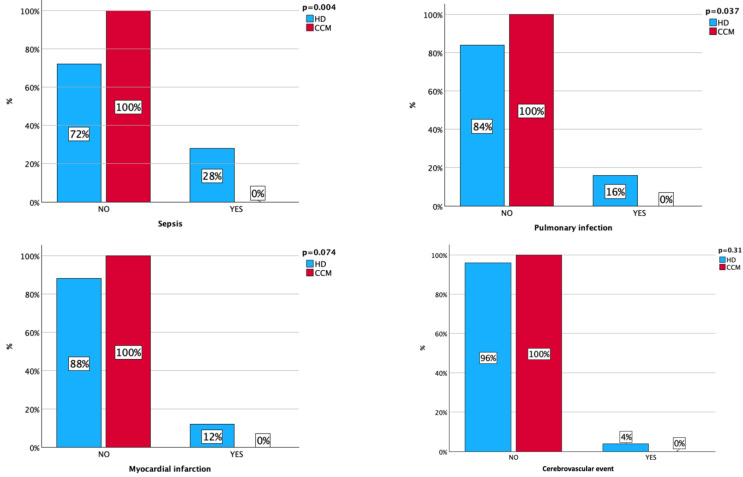
Severe adverse events during the follow-up period.

**Figure 3 nutrients-16-03037-f003:**
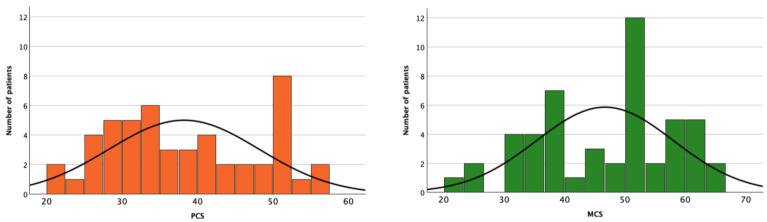
SF-12 questionnaire histogram of the entire population.

**Table 1 nutrients-16-03037-t001:** CCM advantages in elderly ESKD patients with preserved urine output.

Related to Lack of HD Complications	Related to CCM
Avoiding intradialytic hypotension	Slow CKD progression
Avoiding intradialytic cramps	Adequate control of urea retention
Limiting hospital facility access	Adequate control of anemia
Avoiding vascular access thrombosis	Adequate control of calcium–phosphate metabolism
Avoiding vascular access infection	Adequate control of metabolic acidosis
Reducing hospitalization rate	Positive impact of blood pressure control

Footnotes: HD: hemodialysis, CCM: comprehensive conservative management, CKD: chronic kidney disease.

**Table 2 nutrients-16-03037-t002:** Baseline characteristics of the whole population and according to treatment.

	Entire Population	HD	CCM	*p*-Value
Age (years) *	81.5 (±4.5)	80.6 (±4.7)	82.5 (±4.3)	0.15
Female (%)	40	24	56	0.021
CCI 9 10 11 12 13 14	14 28 12 18 12 6	12 28 24 20 12 4	16 28 20 16 12 8	0.98
eGFR * (mL/min)	8.22 (±2.1)	7.95 (±1.7)	8.45 (±2.4)	0.37
Urea * (mmol/L)	28.5 (±9.5)	30.9 (±9.5)	26.1 (±9)	0.07
Hemoglobin * (g/dL)	10.7 (±1.6)	10.6 (±1.3)	11 (±1.6)	0.002
BMI * (Kg^2^/m^2^)	24.4 (± 3.6)	24.7 (±4.06)	24.1 (±3)	0.51
Albumin * (g/L)	3.49 (±0.5)	3.4 (±0.54)	3.6 (±0.54)	0.14
Sodium * (mmol/L)	138 (±4.6)	137.4 (±5.6)	139.8 (±2.9)	0.064
Potassium * (mmol/L)	4.2 (±0.79)	4.2 (± 1.08)	4.3 (±0.5)	0.73
Calcium * (mmol/L)	2.23(±0.18)	2.21 (±0.23)	2.25 (±0.13)	0.45
Phosphate ^ (mmol/L)	1.56 [1.3–1.8]	1.6 [1.4–1.8]	1.48 [1.2–1.7]	0.28

Footnotes: CCI: Charlson Comorbidity Index, eGFR: estimated Glomerular Filtration Rate, BMI: body mass index, * reported as mean (±standard deviation), ^ reported as median [interquartile range].

**Table 3 nutrients-16-03037-t003:** Characteristics of the entire population and of HD and CCM patients at the end of the follow-up period.

	Entire Population	HD	CCM	*p*-Value
Urine output * (cc/d)	1.34 (±0.50)	1.09 (±0.49)	1.6 (±0.37)	<0.001
BMI * (Kg/m^2^)	24.3 (±3.2)	24.5 (±3.7)	24.2 (±2.9)	0.84
Urea * (mmol/L)	24.6 (±7.7)	27 (±8.2)	21.9 (±6.1)	0,01
Albumin *(g/L)	3.7 (±0.5)	3.7 (±0–4)	3.6 (±0.6)	0.54
Hemoglobin (g/dL)	10.9(±1.4)	10.8(±1.5)	11 (±1.2)	0.61
Sodium *(mmol/L)	137.7 (±3.5)	136.8 (±3.5)	138.6 (±3.4)	0.08
Potassium (mmol/L)	4.5 (±0.8)	4.7 (±0.8)	4.4 (±0.7)	0.01
Calcium * (mmol/L)	2.3 (±0.29)	2.2 (±0.19)	2.24 (±0.4)	0.28
Phosphate ^ (mmol/L)	1.37 [1.3–1.7]	1.6 [1.3–1.8]	1.3 [1.1–1.55]	0.03
PTH ^ (ng/L)	286 [172–370]	274 [167.5–399]	288 [178–348]	0.82
Bicarbonate * (mmol/L)	23.6 (±3.2)	25.9 (±0.8)	23.5 (±3.2)	0.31
VAS ^ score	4 [2–6]	6 [2–6]	4 [2–6]	0.09

Footnotes: BMI: body mass index, * reported as mean (±standard deviation), ^ reported as median [interquartile range].

**Table 4 nutrients-16-03037-t004:** PCS and MCS univariate analysis.

	Physical Composite Score			Mental Composite Score		
	B	*p*	95% CI	B	*p*	95% CI
Age (^)	0.003	0.72	−0.14 0.02	0.001	0.98	−0.1 0.017
CCM	0.24	0.001	0.1 0.38	0.15	0.048	0.001–0.3
Female	0.052	0.51	−0.1 0.21	−0.01	0.91	−0.02 0.13
Protein intake	−0.004	0.008	−0.007 −0.00	−0.002	0.26	−0.005 0.001
Urine output	0.25	0.001	0.1 0.4	0.18	0.02	0.028 0.33
BMI	0.01	0.43	−0.02 0.04	0.004	0.77	−0.23 0.3
Hemoglobin	−0.008	0.78	−0.06 0.05	−0.31	0.26	−0.09 0.024
Albumin	−0.06	0.47	−0.21 0.1	0.003	0.96	−0.15 0.16
Urea	−0.009	0.09	−0.02 0.001	−0.006	0.26	−0.02 0.004
Sodium	0.009	0.4	−0.01 0.03	0.013	0.24	−0.01 0.035
Potassium	0.02	0.67	−0.08 0.12	0.004	0.93	−0.97 0.105
Calcium	0.18	0.16	−0.07 0.45	0.024	0.86	−0.24 0.29
Phosphate (^)	−0.14	0.43	−0.49 0.22	−0.31	0.083	−0.65 0.41
PTH (^)	−0.08	0.24	−0.22 0.06	−0.27	0.7	−0.17 0.114
Bicarbonate	0.008	0.66	−0.3 0.04	0.01	0.58	−0.25 0.044
Hospitalization	−0.18	0.024	−0.33 −0.25	−0.1	0.21	−0.26 0.06
VAS score (^)	−0.26	<0.001	−0.4 −0.14	−0.14	0.048	−0.28 −0.001

Footnotes: 95%CI: 95% confidence interval. (^) Normalized covariate.

**Table 5 nutrients-16-03037-t005:** PCS and MCS multivariate stepwise analysis.

	Physical Composite Score			Metal Composite Score		
	B	*p*	95% CI	B	*p*	95% CI
CCM	0.19	0.003	0.07 0.32	0.16	0.03	0.16 0.3
VAS score ^	−0.24	<0.001	−0.36 −0.12	−0.157	0.024	−0.28 −0.02

Footnotes: 95%CI: 95% confidence interval. (^) Normalized covariate.

## Data Availability

The data presented in this study are available on request from the corresponding author.

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
