# Peer review of "The Quality of Life in Elderly Patients in Comprehensive Conservative Management or Hemodialysis: A Case–Control Study in Analogous Basal Conditions"

_nutrients, 2024, doi:10.3390/nu16173037_

Round 1

Reviewer 1 Report

Comments and Suggestions for Authors

Review of the manuscript – nutrients-3178397

Is comprehensive conservative management a better option 2 than hemodialysis for the quality of life of elderly end-stage 3 kidney disease patients? A case-control study with basal comparable health conditions.

By Francesca K. Martino et al

The evaluated manuscript presents an original article. The authors aimed to assess the differences in the quality of life (QoL) among patients with chronic kidney disease, specifically those with End-Stage Kidney Disease (ESKD). The study divided participants into two groups: one receiving conservative management (CM; non-dialyzed) and the other undergoing hemodialysis (HD). Both groups were similar in age, comorbidity index, and overall health. The secondary objective was to evaluate how various clinical factors—such as age, comorbidity index, ESKD management, uremia control, pain perception, and hospitalization—could influence QoL. The authors concluded that patients in the conservative management group experienced significantly better physical and mental well-being compared to those in the hemodialysis group. They found that conservative management positively impacts QoL more than hemodialysis. Additionally, the conservative management group exhibited lower rates of severe infections, better control of urea retention, sodium, potassium, and phosphate levels, as well as higher urine output. The results of the study are surprising; while one might expect an improvement in well-being among older patients with ESKD in the conservative management group due to fewer limitations associated with forgoing dialysis, the more pronounced improvement in selected clinical parameters compared to the hemodialysis group is noteworthy.

Reading the manuscript raises several questions and concerns:

1. Throughout the text (including tables), there are two abbreviations used for the group of patients undergoing comprehensive conservative treatment: This inconsistency needs to be addressed for clarity and uniformity.

2. The Introduction could clearly outline the indications for implementing CCM – e.g. in the form of a separate table.

3. Although CCM is a recognized alternative to renal replacement therapy, it is important to elaborate on the controversies surrounding the choice of this procedure in both the Introduction and Discussion sections. Additionally, a brief characterization of the differing conclusions regarding survival time and the clinical consequences of implementing CCM compared to HD and Continuous Ambulatory Peritoneal Dialysis (CAPD) should be included.

4. In addition to nutritional modifications, did the CCM group, as well as the HD group, receive any additional therapies? What about other aspects of conservative treatment for CCM patients, such as the management of anemia, bone disorders, edema, lactic acidosis, hyperkalemia, pain, itching, anxiety, and depression?

5. I do not fully understand the time frame of the study. The CCM program commenced on September 30, 2022, while patients who initiated CCM or HD were recruited between April 1, 2022, and March 31, 2023. What about the period for the earliest patients enrolled in the study prior to September 30, 2022?

6. The study population exhibited a significant difference in the percentage of male and female subjects within the analyzed HD/CCM groups. While the CCM group comprised nearly equal numbers of male and female subjects, the HD group included only about 25% female patients. Are there potential factors, such as hormones, that could influence the results obtained? Furthermore, this limitation should be addressed in the discussion and explicitly listed in the section.

7. Statistical Analysis – It is important to note in this subsection that when dealing with data that follows a normal distribution, both the mean and standard deviation can be effectively analyzed. In contrast, when the data does not conform to a normal distribution, it becomes essential to interpret the results using the median and the range or quartiles.

8. Line 182: However, CCM patients were more likely to be female (76 versus 52, p=0.02). How is this possible when there were only 50 patients in total in the study?

9. Lines 190-191 (...) At the end of the observational period, CCM patients with chronic kidney disease (CM) exhibited significantly lower levels of urea (p=0.01), sodium (p=0.03), potassium (p=0.01), and phosphate (p=0.049). However, these findings are inconsistent with the results presented in Table 2, where no statistical significance was indicated for sodium. Additionally, the p-value for intergroup differences in phosphate differs from the reported value.

10. Line 197 – “during follow-up…” – What was the specific duration of the follow-up period?

11. Figure 1 presents intergroup differences at each time point, which warrants further analysis. The same observation applies to Figure 2.

12. Chapter 6 "Patents" – This chapter contains information about patients included in the study. This fragment should be moved to Chapter 2 Materials and Methods (e.g., as subchapter 2.1.).

13. The authors themselves indicate that the results obtained contradict many other studies, while also acknowledging numerous limitations and stating their intention to reanalyze the data. Perhaps it would be beneficial to emphasize more strongly (e.g., in the title) that the presented content represents "preliminary results

14. Technical Note: Figures 1-3 are of poor quality (low resolution) and do not allow for precise reading of information, such as the scale.

Author Response

Dear Reviewer,

Thank you for your interest and thorough feedback. Your suggestions and comments have been valuable in enhancing our manuscript and effectively emphasizing the results of our study. The rigorous assessment process has played a crucial role in elevating the overall quality of our work. We acknowledge any shortcomings in our study reporting and appreciate your understanding.

Comment 1 "Throughout the text (including tables), there are two abbreviations used for the group of patients undergoing comprehensive conservative treatment: This inconsistency needs to be addressed for clarity and uniformity"

We have changed as required.

Comment 2 "The Introduction could clearly outline the indications for implementing CCM – e.g. in the form of a separate table"

Thanks for the suggestion, which establishes the study's background and underlines its importance. According to the other Reviewer's suggestions, we have added a table in Table 1.

Comment 3 "Although CCM is a recognized alternative to renal replacement therapy, it is important to elaborate on the controversies surrounding the choice of this procedure in both the Introduction and Discussion sections. Additionally, a brief characterization of the differing conclusions regarding survival time and the clinical consequences of implementing CCM compared to HD and Continuous Ambulatory Peritoneal Dialysis (CAPD) should be included."

In the introduction, we added a paragraph about the controversies in the possible treatment choice for ESKD.

Comment 4 "In addition to nutritional modifications, did the CCM group, as well as the HD group, receive any additional therapies? What about other aspects of conservative treatment for CCM patients, such as the management of anemia, bone disorders, edema, lactic acidosis, hyperkalemia, pain, itching, anxiety, and depression?"

All CCM and HD patients received supportive treatment related to CKD. We did not report the pharmacological treatment because both groups potentially had the same opportunity to have erythropoietin, potassium and phosphorus binder, VitD analogs, Calcium mimetics, sodium bicarbonate, diuretics, and antihistamine treatment. Thus, considering the retrospective analysis of our report and the lack of proof of treatment adherence, we should have reported these results. However, we reported a statement regarding supportive therapy in a methods paragraph.

We did not detect pharmacological treatment for anxiety and depression. However, considering your comment, we reported this lack in the study's limits.

Comment 5 "I do not fully understand the time frame of the study. The CCM program commenced on September 30, 2022, while patients who initiated CCM or HD were recruited between April 1, 2022, and March 31, 2023. What about the period for the earliest patients enrolled in the study prior to September 30, 2022?"

In our department, CCM was born in September 2022. Before September 2022, all patients who achieved ESKD were addressed to RRT (hemodialysis or peritoneal dialysis). After September 2022, all patients over 75 who reached ESKD could choose between dialysis or CCM. So, to obtain an adequate sample size of 25 patients for groups, we enrolled all patients who started HD between April 2022 and March 2023 and all patients who began CCM from September 2022 to March 2023. Obviously, after CCM was introduced, the number of patients who chose HD decreased significantly. 

We revised the method's paragraph based on your comment.

Comment 6 "The study population exhibited a significant difference in the percentage of male and female subjects within the analyzed HD/CCM groups. While the CCM group comprised nearly equal numbers of male and female subjects, the HD group included only about 25% female patients. Are there potential factors, such as hormones, that could influence the results obtained? Furthermore, this limitation should be addressed in the discussion and explicitly listed in the section."

Considering the age of our patients, it is hard to attribute a significant difference related to hormonal conditions. This difference could be associated with the different propensities among men and women in choosing ESKD treatment, as Morton et al. reported. However, we added a statement in the discussion.

Comment 7 "Statistical Analysis – It is important to note in this subsection that when dealing with data that follows a normal distribution, both the mean and standard deviation can be effectively analyzed. In contrast, when the data does not conform to a normal distribution, it becomes essential to interpret the results using the median and the range or quartiles."

We do not fully understand this comment because it is what we did, as we reported in the statistical analysis. We wrote the paragraph again to be more detailed.

Comment 8 "Line 182: However, CCM patients were more likely to be female (76 versus 52, p=0.02). How is this possible when there were only 50 patients in total in the study?"

We apologize for this typo; the correct values were reported in Table 2.

Comment 9 "Lines 190-191 (...) At the end of the observational period, CCM patients with chronic kidney disease (CM) exhibited significantly lower levels of urea (p=0.01), sodium (p=0.03), potassium (p=0.01), and phosphate (p=0.049). However, these findings are inconsistent with the results presented in Table 2, where no statistical significance was indicated for sodium. Additionally, the p-value for intergroup differences in phosphate differs from the reported value."

We apologize for the incongruence; there was an issue in the main text; we corrected the paragraph according to the actual value reported in Table 3.

Comment 10 "Line 197 – "during follow-up…" – What was the specific duration of the follow-up period?"

The observational period was six months. We added this note in the text.

Comment 11 "Figure 1 presents intergroup differences at each time point, which warrants further analysis. The same observation applies to Figure 2."

Figure 1 represents the number of hospitalizations in the six months of observation, while figure 2 shows the rate of sepsis, severe pulmonary infections, myocardial infarction, and cerebrovascular events in the same period. We changed the figures and the text to clarify the message better.

Comment 12" Chapter 6 "Patents" – This chapter contains information about patients included in the study. This fragment should be moved to Chapter 2 Materials and Methods (e.g., as subchapter 2.1.)"

Chapter 6 is an optional paragraph suggested by the journal, and it appears unnecessary in our context. We deleted it.

Comment 13. "The authors themselves indicate that the results obtained contradict many other studies, while also acknowledging numerous limitations and stating their intention to reanalyze the data. Perhaps it would be beneficial to emphasize more strongly (e.g., in the title) that the presented content represents "preliminary results."

As usual in the discussion, we compared our results with others. Specifically, one study moved in our direction; three studies had a nonsignificant difference in QoL between the CCM and HD groups, and only one study showed a better quality of life in hemodialysis patients. In all these studies, the basal conditions were significantly different in age and comorbidities. Almost all studies that did not achieve a significant difference in QoL described a significant discrepancy in basal conditions. Thus, the older and higher comorbid patients in CCM could have experienced a worse quality of life independently from ESKD treatment than younger and less comorbid patients. Furthermore, some of these studies compared CCM and RRT (hemodialysis and peritoneal dialysis); obviously, the presence of home-based treatment as peritoneal dialysis in the control group can dilute the advantage in the quality of life of CCM concerning HD. We have reconsidered the related paragraphs in the discussion regarding these points.

Comment 14 "14. Technical Note: Figures 1-3 are of poor quality (low resolution) and do not allow for precise reading of information, such as the scale"

Thanks for the note, and sorry for the inconvenience; we have changed the figures.

Reviewer 2 Report

Comments and Suggestions for Authors

The authors studied a very relevant topic faced by nephrologists on a daily basis whether to manage the elderly ESKD patients conservatively or with dialysis. I would like to know the reason why the authors specifically chose in-center hemodialysis patients versus those on home hemodialysis and peritoneal dialysis. 

I have several other comments. 

1. Extensive edition of English language and grammar is required throughout the manuscript. 

2. I think the title is too long. It is better to have a concise title. A full stop cannot appear on the title. 

3. Abstract: Be consistent in using either end stage kidney disease or end stage renal disease. Use one term only and not both. Better to abbreviate ESKD on its first use and then only use ESKD later throughout the manuscript. 

4. Abstract: VAS? What does it stand for? Expand on its first use

5. Abstract: should be mental composite score (MCS) and not mental composite (MCS) 

6. Abstract: What is FCS? 

7. Introduction: I think it is too long. Shorten it and keep to at list short 3 paragraphs only

8. Introduction: need extensive English language edition, such as instead of after the HD begins, use "after initation of HD"

9. Introduction: 2nd paragraph; remove Or after ref. 13

10. Use first, second and third, instead of "firstly", "secondly" and "thirdly"

11. Instead of "limiting access to hospital facilities", use "without the need to access hospital facilities" so that the context is better understood

12. After already defining abbreviation, such as QOL, do not use "Quality of Life" again later

13. Materials and Methods: What is UO?; ESKD Is expanded again (already expanded before). Why the eGFR cut off was < 12 and not < 15 (the definition of ESKD). I think 6 month time is too short to assess the QOL. Why home HD patients were excluded? 

14. CM and CCM are used at various places- use one term consistently

15. PTH: parathormone?, it should be parathyroid hormone

16. Table 1: Is age reported as mean or median- specify in the table

17. Result: The most concerning of the manuscript is how did authors ensure that the HD patients followed a free-protein diet?? Is there such thing as 100% free protein diet. Specify the list of food that are 100% protein diet and even if so, what measures were done at patient's home to ensure that they indeed did not take protein diet at home when they were not being supervised. This can totally alter the result of the manuscript. 

18. Patents? Not sure what authors meant? It is stated after conclusion. It seems an inclusion and exclusion criteria. 

Comments on the Quality of English Language

Extensive English language correction is required 

Author Response

Dear Reviewer,

We want to express our sincere gratitude for your invaluable comments and suggestions. They have proven to be incredibly helpful for us. Your timely evaluation of both the content and form has undoubtedly enhanced the quality of our work. We regret any oversights in our study reporting and appreciate your understanding.

Comment. I would like to know why the authors specifically chose in-center hemodialysis patients versus those on home hemodialysis and peritoneal dialysis. 

We decided to compare conservative management and hemodialysis patients for two different reasons.  Hemodialysis is the most prevalent treatment modality for patients needing renal placement treatment, as reported in the ERA Registry Annual Report 2021 and USRDS [doi: 10.1093/ndt/gfae040.], especially in patients over 75 years old [doi: 10.1093/ckj/sfad281]. Home hemodialysis is performed in younger and low comorbid patients [doi:10.1002/14651858.cd009535.pub3]. Furthermore, home dialysis barriers such as patients/caregivers fear, lack of space, lack of home-based support, and limited support for staff and patients limit the diffusion of peritoneal dialysis and home hemodialysis. These issues are pronounced in elderly patients. So, we choose in-center hemodialysis to have comparable basal conditions.  

Comment :Extensive edition of English language and grammar is required throughout the manuscript.

We provide an extensive revision of English.

Comment 2: I think the title is too long. It is better to have a concise title. A full stop cannot appear on the title. 

We thank you for your suggestion. We changed the title to "Quality of Life in elderly patients in Comprehensive Conservative Management or Hemodialysis: A case-control study in analogous basal conditions"

Comment 3. Abstract: Be consistent in using either end stage kidney disease or end stage renal disease. Use one term only and not both. Better to abbreviate ESKD on its first use and then only use ESKD later throughout the manuscript. 

We changed as suggested. 

Comment 4. Abstract: VAS? What does it stand for? Expand on its first use.

We changed as suggested. 

Comment 5. Abstract: should be mental composite score (MCS) and not mental composite (MCS).

We changed as suggested. 

Comment 6. Abstract: What is FCS?  

It is a typo mistake instead of PCS, thanks

Comment 7. Introduction: I think it is too long. Shorten it and keep to at list short 3 paragraphs only

We changed as suggested, according to another reviewer's comment.

Comment 8. Introduction: need extensive English language edition, such as instead of after the HD begins, use "after initation of HD"

We changed, as suggested.

Comment 9. Introduction: 2nd paragraph; remove Or after ref. 13

We changed, as suggested.

Comment 10. Use first, second and third, instead of "firstly", "secondly" and "thirdly"

We changed, as suggested.

Comment 11. Instead of "limiting access to hospital facilities", use "without the need to access hospital facilities" so that the context is better understood

We changed, as suggested.

Comment 12. After already defining abbreviation, such as QOL, do not use "Quality of Life" again later

We changed, as suggested.

Comment 13. 

Materials and Methods: What is UO?

We corrected the affiliation in the text; 

ESKD Is expanded again (already expanded before). 

We changed, as suggested

Why the eGFR cut off was < 12 and not < 15 (the definition of ESKD).

We focus on patients with eGFR <12 ml/min to reduce the gap in kidney function between the CCM and HD; as we showed in Table 2, there was no significant difference between cases and controls.

I think 6 month time is too short to assess the QOL.

In this study, we selected patients who had been stable on HD or CCM for 6 months before the start of the observation period. We monitored these patients for 6 more months to identify any potential complications during treatment. Therefore, the SF-12 results reflect the patients' perception of their quality of life after one year of treatment. Additionally,other studies, including those by van Loon et al. and So et al., also used similar timing to evaluate quality of life.

Why home HD patients were excluded

Finally, regarding home HD, see the answer to comment 1.

Comment 14. CM and CCM are used at various places- use one term consistently.

We changed, as suggested.

Comment 15. PTH: parathormone?, it should be parathyroid hormone,

We changed, as suggested.

Comment 16. Table 1: Is age reported as mean or median- specify in the table.

We clarified, as suggested.

Comment 17Result: The most concerning of the manuscript is how did authors ensure that the HD patients followed a free-protein diet?? Is there such thing as 100% free protein diet. Specify the list of food that are 100% protein diet andeven if so, what measures were done at patient's home to ensure that they indeed did not take protein diet at home when they were not being supervised. This can totally alter the result of the manuscript. 

We apologize for the unfortunate confusion caused by the wording, which conveys the wrong message. As detailed in the methods, HD patients followed a free diet. Specifically, based on monthly blood tests, they received recommendations to manage potassium or phosphate levels better.

Comment 18Patents? Not sure what authors meant? It is stated after conclusion. It seems an inclusion and exclusion criteria.

Chapter 6 is an optional paragraph suggested by the journal, and it appears unnecessary in our context. We deleted it.

Round 2

Reviewer 1 Report

Comments and Suggestions for Authors

Thank you for sending the revised version of the manuscript. The authors have responded to my comments in detail and have taken into account my suggestions. I believe that the manuscript can be published.

Author Response

Dear reviewer,

Thank you for your comment on our revised submission. We appreciate your feedback.

Best regards, Francesca Martino

Reviewer 2 Report

Comments and Suggestions for Authors

The authors have modified the manuscript as per suggestions. Minor comments to the author: Please revise the last statement in conclusion as follows: In elderly patients (remove comorbid), CCM may allow a better perception of QoL and hence may be a best option as opposed to HD. 

Author Response

Dear Reviewer,

I am sincerely grateful for your insightful feedback on our revised submission. Your constructive comments have significantly influenced our work. We have duly modified the closing sentence.

Best regards, Francesca Martino